# Breast Cancer in the Tissue of the Contralateral Breast Reduction

**DOI:** 10.3390/cancers16030497

**Published:** 2024-01-24

**Authors:** Zoë M. A. Kuijlaars, Nadine S. Hillberg, Loes Kooreman, Carmen A. H. Severens Rijvers, Shan Shan Qiu

**Affiliations:** 1Department of Plastic, Reconstructive and Hand Surgery, Maastricht University Medical Center+, 6229 HX Maastricht, The Netherlands; z.kuijlaars@student.maastrichtuniversity.nl (Z.M.A.K.); nadine.hillberg@mumc.nl (N.S.H.); 2Department of Pathology, Maastricht University Medical Center+, 6229 HX Maastricht, The Netherlands; loes.kooreman@mumc.nl (L.K.); carmen.rijvers@mumc.nl (C.A.H.S.R.); 3GROW School for Oncology and Reproduction, Maastricht University Medical Center+, 6229 HX Maastricht, The Netherlands

**Keywords:** breast cancer, breast reduction, contralateral breast cancer

## Abstract

**Simple Summary:**

Breast cancer is the most prevalent malignancy among women globally. Early diagnosis and treatment improvements are leading to a growing population of survivors. This has increased the risk of developing contralateral breast cancer (CBC), a distinct occurrence in the opposite breast of a previously diagnosed patient. The treatment options for breast cancer are often mastectomy or lumpectomy. Patients with lumpectomy frequently undergo a contralateral reduction procedure to achieve more symmetry after the primary breast cancer surgery. The reduction specimen is usually routinely examined by pathology to check for malignancies. The excision in pieces and the absence of specific markers or ink make an examination of tumor size and margin status more challenging, impacting treatment decisions. A new protocol introduced in July 2022 seeks to improve diagnostic precision and treatment planning via excision in toto and by marking and inking excised reduction tissue to examine and treat potential CBC more effectively.

**Abstract:**

Breast cancer is the most prevalent malignancy among women worldwide, and the increasing number of survivors is due to advances in early diagnosis and treatment efficacy. Consequently, the risk of developing contralateral breast cancer (CBC) among these survivors has become a concern. While surgical intervention with lumpectomy is a widely used primary approach for breast cancer, post-operative breast asymmetry is a potential concern. Many women opt for symmetrizing reduction procedures to improve aesthetic outcomes and quality of life. However, despite careful radiological screening, there is a chance of accidentally finding CBC. To address this, tissue excised during symmetrizing surgery is examined pathologically. In some cases, CBC or in situ lesions have been incidentally discovered in these specimens, prompting a need for a more thorough examination. Resection in pieces and the absence of surgical marking and pathological inking of the margin have made it challenging to precisely identify tumor location and assess tumor size and margin status, hampering adjuvant treatment decisions. A new protocol introduced in July 2022 aims to enhance the precision of CBC diagnosis, allowing for tailored treatment plans, including re-excision, systemic adjuvant therapy, or radiation therapy.

## 1. Introduction

Breast cancer stands out as the most prevalent malignancy among women worldwide. The increasing number of breast cancer survivors can be attributed to advancements in early diagnosis and improvements in treatment efficacy [1,2]. Consequently, an expanding cohort of women is exposed to the risk of developing contralateral breast cancer (CBC). Within the domain of breast cancer research and treatment, CBC emerges as a significant focal point. This terminology denotes the manifestation of a novel and distinct breast cancer occurrence in the contralateral breast of a patient previously diagnosed with breast cancer of the other breast. CBC carries an adverse prognosis, characterized by an augmented incidence of distant metastases and elevated mortality rates linked to breast cancer [3].

Whenever primary surgery for breast cancer treatment is indicated, patients typically receive recommendations for either mastectomy or breast-conserving surgery. Nevertheless, both options entail the potential for post-operative breast asymmetry. Consequently, a substantial number of women opt for symmetrizing reduction procedures, which result in a more harmonious aesthetic outcome and contribute to an enhanced quality of life [4].

For most patients with primary breast cancer, the risk of developing distant metastases surpasses the risk of developing contralateral breast cancer. Approximately 10–12% of women treated for primary breast cancer experience distant recurrence during a mean follow-up of slightly over 5 years [5]. However, the risk of contralateral breast cancer is 1.3 to 1.9 times higher than the risk of primary breast cancer in the general population [3,6] and reaches up to a 10.5% cumulative incidence in 20 years (3).

Consequently, the tissue excised during symmetrizing surgery undergoes pathological examination to discern any potential signs of cancer. This specimen was previously resected in multiple fragments and not marked in any specific way. On several occasions, CBC or in situ lesions within this specimen were incidentally discovered in our institution. This serendipitous discovery prompted us to reconsider the entire protocol from surgical removal to examination at the pathology department. The absence of marking and inking and removal in pieces made it challenging to identify, precisely, the tumor’s location and ascertain whether it had been completely excised. Secondly, it was sometimes challenging to define tumor size, which is important for staging and defining the need for adjuvant (systemic) therapy depending on the tumor’s characteristics; many patients were required to undergo either mastectomy or whole-breast radiation therapy. By introducing a new protocol on 20 July 2022, our objective was to enhance our ability to provide detailed information regarding the potential CBC, such as tumor size and the margin status. This improved precision in diagnosis may allow for the adjustment of treatment plans accordingly, potentially involving re-excision, chemotherapy, targeted therapy, or a boost of radiation therapy.

### Breast Cancer Treatment Guidelines

The treatment approach for breast cancer depends on the tumor type and is discussed within a Multidisciplinary Oncology Board (MOB).

The importance of adjuvant therapy varies depending on individual patients and tumor characteristics, with certain factors, like tumor size and margin status, being more challenging to ascertain in cases of non-oncologic resection without protocolized removal and specimen handling.

Tumor size primarily influences the determination of chemotherapy and targeted therapy. The assessment of tumor grade or receptor status is generally not significantly affected by the mode of resection. Determining surgical margins is crucial for considering additional surgery and radiotherapy, possibly with a boost.

Follow-up is indicated for other diagnoses such as lobular carcinoma in situ (LCIS). This excludes the pleomorphic type of LCIS as it is treated the same as DCIS [1]. For atypical ductal hyperplasia (ADH), no intervention is recommended.

## 2. Materials and Methods

A retrospective cohort study was performed examining patients who received a symmetrizing breast reduction, due to previous breast cancer treatment in the contralateral breast, between January 2018 and December 2022 at the Maastricht University Medical Center+ (MUMC+) in the Netherlands.

The inclusion criteria for this analysis involved patients in whom (pre)malignancies were identified within the breast reduction tissue.

### 2.1. Data Management

Data were recorded from the patient’s medical records concerning oncological, radiological, and pathological parameters, with baseline characteristics: age, age at the time of first and second diagnoses, body mass index (BMI), history of intoxication, and medication history. These variables were deemed essential for a comprehensive understanding of breast cancer dynamics and their potential influence on outcomes.

The study also scrutinized initial cancer diagnosis details, pathology reports, and treatment modalities, serving as crucial context for comprehending the patients’ breast cancer journeys. Concerning the pathology, we collected information on histological subtype, tumor diameter, grade, lymph vascular space invasion, hormonal and Her2 receptors, and margin status. Concerning radiology, we collected information on BI-RADS classification, tumor localization, and diameter. The BI-RADS classification, which stands for “Breast Imaging Reporting and Data System”, is a standardized system used in radiology and mammography to describe and report findings related to breast imaging [7]. Data were collected for the primary and secondary diagnosis.

### 2.2. Surgical Technique

All patients underwent contralateral breast reduction surgery for symmetry using the Wise pattern technique. This is the most common technique used in breast reduction surgery [8]. The surgical protocol was changed on 20 July 2022 as resections were made in toto and marking of the excised tissue was conducted cranially (Figure 1). This consensus started since that date and all the plastic surgeons involved in this procedure followed it accordingly. Before this protocol, the resection of the breast tissue could be conducted in toto or in small pieces, and no marking was requested.

### 2.3. Pathology: Protocol and Methods

The pathology protocol changes involved inking the reduction preparations of patients with a previous breast malignancy and in patients above the age of 50 (Figure 2). Additionally, the transected slices at grossing are saved in their original order. In case of an accidental finding at microscopy, it is possible to identify the selection site of the original piece obtained for microscopy. Additional sampling can be conducted to reconstruct tumor size and margin status. (Figure 2).

## 3. Results

### 3.1. Participants

Between January 2018 and December 2022, 648 patients had a symmetrizing contralateral breast reduction. In 10 of them, a (pre-)malignancy was found in the breast reduction tissue. These 10 patients were included for further investigation. The incidence rate was 1.5% in 5 years.

### 3.2. Patient Demographics and Clinical Characteristics

For baseline patient characteristics, see Table 1.

### 3.3. Main Results

#### 3.3.1. Pathological Information

Table 2 shows the main pathological information regarding the primary tumor and the secondary tumor found at the contralateral breast.

#### 3.3.2. Radiological and Surgical Information

Regarding the primary tumor, a BI-RADS classification of 5 was assigned to all patients. Of these, eight opted for a lumpectomy, and two chose to undergo mastectomy. Among these patients, three underwent axillary clearance, while six opted for a sentinel node procedure. Furthermore, five patients received neoadjuvant chemotherapy. Additionally, two patients received adjuvant chemotherapy, and six received adjuvant radiotherapy. The median diameter of the primary tumor, as ascertained from mammographic imaging, was measured at 24 mm (Table 3).

As for the secondary tumor, a BI-RADS classification of 2 was assigned to nine out of ten patients, and one patient received a classification of 1. Based on the pathology, two patients elected to undergo mastectomy upon diagnosis, and one of them received adjuvant radiotherapy. It is noteworthy that none of these secondary tumors were diagnosed in the pre-operative imaging screening (Table 3).

#### 3.3.3. Follow-Up

In this study, all patients underwent yearly follow-up breast examinations for 5 years following their initial tumor diagnosis, which included mammograms of the both affected breast and of the contralateral breast in the case of breast conserving surgery and a check-up at the oncologist. Four patients stood out in this regard, as they completed their 5-year follow-up but were subsequently diagnosed with CBC. Notably, their mammograms consistently showed benign findings (BI-RADS 2) over the entire 5-year follow-up period. One of these patients developed CBC 29 years after concluding the 5-year follow-up, while the other patients received the CBC diagnosis 12, 3, and 1 year(s) after completing their 5-year follow-up.

Furthermore, two additional patients completed their 4-year follow-up, maintaining a BI-RADS 2 classification throughout this time before eventually receiving a CBC diagnosis. Similarly, four patients completed their 3-year follow-up, with BI-RADS 2 findings consistently reported during each examination before eventually being diagnosed with a second case of breast cancer.

However, it is worth noting that for one patient, the radiological data were unavailable for the primary tumor. The first diagnosis was in 1992, giving us no BI-RADS classification or other radiological information.

Before every breast reduction surgery, a mammogram of the breast that receives the symmetrical reduction is mandatory, and it should be no older than 6 months to rule out any malignancies. The pre-operative breast reduction protocol changed on 1st October 2023, before that date, the one-year-old mammogram has been sufficient whereas after that date, the mammogram cannot be older than 6 months [9].

#### 3.3.4. Treatment Plan after the Secondary Diagnosis

For six out of ten patients, there was no alteration in their treatment plan, nor was it recommended by the MOB. These patients were diagnosed with either LCIS or ADH. The clinical management in these cases aligned with established guidelines, which advocate for radiological follow-up over a period of five years.

Conversely, the four patients who received a diagnosis of DCIS, invasive lobular carcinoma, or invasive carcinoma NST (of non-special type) underwent comprehensive discussions within the MOB. In each of these cases, the recommendations encompassed either mastectomy or lumpectomy, an increased dosage of radiotherapy, or diligent follow-up, all depending on margins. One of the patients, diagnosed with invasive carcinoma NST, had clear tumor margins and was either advised to have adjuvant radiotherapy or a wait-and-see policy. She opted for the latter. The second patient, diagnosed with invasive carcinoma NST, was advised to have a mastectomy or radiotherapy based on the unclear margins. She underwent a mastectomy. Both diagnoses were made before the protocol changed, leaving the second patient with no options.

It is worth noting that one patient with DCIS received this diagnosis after the protocol modification, affording them the option of choosing a lumpectomy instead of a mastectomy. The margins of the tumor were not radical, but because of the changed protocol, localization of the tumor was made, and a lumpectomy was possible. Nevertheless, due to concerns about the potential recurrence of breast cancer and the breast reconstruction possibilities, the patient opted for a mastectomy with autologous tissue reconstruction.

## 4. Discussion

In the context of this study, CBC was observed with an incidence of 1.5% in the population under investigation at our institution. This incidence also includes cases of pre-malignancies. However, when considering only malignancies, excluding cases of LCIS, the incidence rate stood at 0.77%. This is slightly higher than the 0.44% of the general CBC incidence [3]. The occurrence of invasive carcinoma in general reduction mammoplasties can differ between 0.06 and 5%, and the larger cohorts show an average incidence ranging from 0.06% to 0.38%. The incidence of CBC is twice as high [10]. Interestingly, out of the 10 patients who had (pre)malignant conditions, they underwent regular follow-up examinations with imaging for 5 years after the primary diagnosis. During these follow-ups, their BIRADS stages consistently indicated 1 or 2. This suggests that without the symmetrizing breast reduction surgery, the diagnosis of CBC might have been delayed or never made.

The introduction of the revised protocol has significantly broadened the range of available treatment options for patients who have received a breast cancer diagnosis, all while maintaining oncologic safety. The incorporation of complete specimen excision and marking has notably improved the precision of tumor localization. This, in turn, has opened the possibility of considering radiotherapy and surveillance as alternatives to immediate surgical intervention. The utilization of specimen marking simplifies the tumor localization process, providing a more comprehensive assessment of the tumor’s size. This information is crucial for guiding adjuvant systemic therapy. Without specimen marking, accurately pinpointing the tumor’s exact location becomes challenging, leaving questions about margin status. In such cases, mastectomy or comprehensive chest radiation may become the only viable treatment option. When we reviewed our patient cohort, it is worth mentioning that only one individual had DCIS identified in the resected specimen following the protocol amendment. She chose to undergo a mastectomy, primarily due to concerns about potential disease recurrence. This patient represents the only case in our study that truly benefited from the protocol change. However, in the meantime, we have had referral cases from centers without a specific protocol and encountered similar problems, stressing the need for a wider implementation.

It is important to note that the financial impact of specimen marking and inking is minimal, effectively eliminating any potential financial burden. Both marking and inking procedures are brief, requiring only a minimal amount of time during the surgical procedure and pathology specimen handling, respectively. Moreover, the removal of the specimen in toto does not alter the aesthetic outcome.

Upon changing the breast cancer treatment protocol, it becomes evident that there are relatively few consequences for specific categories including LCIS. LCIS is not classified as cancer but as a pre-malignancy. The cumulative incidence of breast cancer following a diagnosis of LCIS is approximately 1–2% per year, resulting in a relative risk that is 8–10 times higher than the general population. According to Dutch guidelines, a 5-year follow-up is sufficient in these instances [1,11]. Since the diagnosis of LCIS does not involve any physical treatment options, psychological support might be a topic to discuss after the diagnosis. Currently, in North America, the standard of care includes the screening of distress symptoms in cancer patients at all stages of their illness. The high prevalence of distress symptoms specific to the diagnosis of breast cancer indicates that healthcare providers should give increased attention to these symptoms. Although LCIS is not a malignancy, it does lead to uncertainty and fear of breast cancer. Early screening and the provision of interventions for distress can enhance adherence to improve quality of life and be a form of treatment [12,13].

Concerns have arisen regarding the potential impact of pre-operative mammograms on the accuracy of predicting malignancies in the reduction specimen. This has led to discussions about the necessity of conducting a more comprehensive pre-operative mammogram assessment. However, upon closer scrutiny, it becomes apparent that smaller abnormalities, such as LCIS and ADH, often go unnoticed on mammograms. Consequently, the outcomes of these specific diagnoses remain unaffected by the pre-operative mammogram.

In general, one of the most relevant benefits of this protocol change is the gain of shared decision-making. Patients have more treatment options available, giving them a higher level of autonomy [14].

## 5. Conclusions

The revised breast cancer contralateral reduction protocol has broadened therapeutic choices by providing better tumor assessment through resection in toto, excision specimen marking, and inking. This enhances oncological safety. The financial impact is minimal. Overall, the revised protocol improves treatment flexibility and gains shared decision-making. Thereby, this article creates more awareness regarding the risk of contralateral breast cancer in reduction specimens. We highly suggest that every plastic surgery and pathology department introduce this new protocol as standard care.

## Figures and Tables

**Figure 1 cancers-16-00497-f001:**
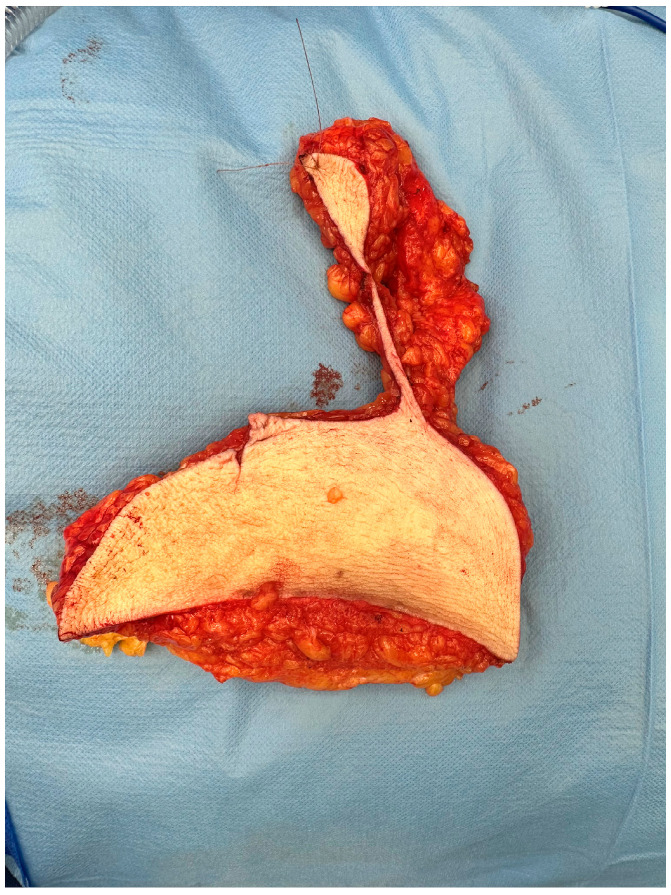
Reduction specimen marked cranially.

**Figure 2 cancers-16-00497-f002:**
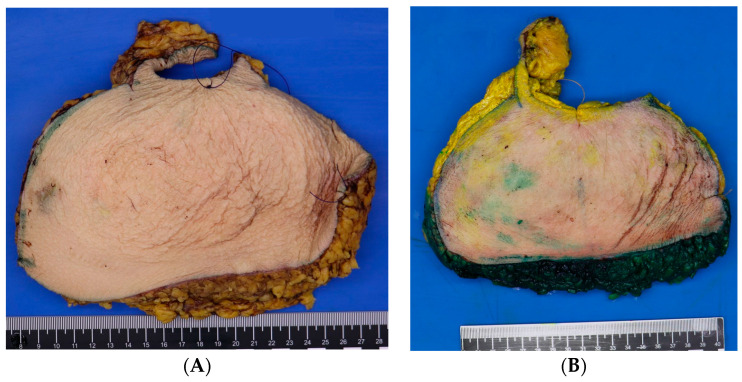
(**A**) Specimen resected in toto with cranial and medial markings. (**B**) Specimen resected in toto with cranial and medial markings, displayed after inking at the pathology department. Yellow was used for the ventrocranial zone and green for the ventrocaudal zone.

**Table 1 cancers-16-00497-t001:** Patient and clinical characteristics.

Characteristic	N = 10
Age at first diagnosis; Median ± IQR ^1^, years	46.5 ± 15
Age at second diagnosis; Median ± IQR, years	59 ± 16
Interval between diagnoses; Median ± IQR, years	4 ± 13
BMI; Median ± IQR, years	27.19 ± 6.43

^1^ Inter quartile range.

**Table 2 cancers-16-00497-t002:** Pathological information of the primary and secondary tumor.

	Primary Tumor N = 10				Secondary Tumor N = 10				
	Type	Grade	Receptor	Size (mm)	Type	Grade	Receptor	Size (mm)	Margin
1	DCIS ^1^	3	ER+ ^2^ (100%)/PR+ ^3^ (0%), HER2− ^4^	25	DCIS	2	N/A	18	Positive
2	DCIS	3	ER+ (100%)/PR+ (25%), HER2−	13	DCIS	2	N/A	3	Negative
3	DCIS	2	ER/PR+, HER2−	41	LCIS ^5^	N/A	N/A	N/A	Negative
4	DCIS	2	ER+ (100%)/PR+ (95%), HER2−	12	LCIS	N/A	N/A	N/A	Negative
5	DCIS	2	ER+ (100%)/PR+ (100%), HER2−	15	NST ^6^	N/A	ER/PR+	1.5	Positive
6	NST	3	Triple negative	37	LCIS	N/A	N/A	N/A	Negative
7	NST	3	Triple negative	53	LCIS	N/A	N/A	N/A	Negative
8	NST	2	ER+ (100%)/PR+ (25%), HER2−	11	NST	2	ER/PR+, HER2−	6	Positive
9	NST	2	ER+ (95%)/PR+ (85%), HER2 amplified	24	ADH ^7^	N/A	N/A	N/A	Negative
10	NST	2	ER− ^8^	42	ADH	N/A	N/A	N/A	Negative

^1^ Ductal carcinoma in situ; ^2^ Estrogen receptor; ^3^ Progesteron receptor; ^4^ Human epidermal growth factor receptor 2; ^5^ Lobular carcinoma in situ; ^6^ Invasive carcinoma of no special type; ^7^ Atypical ductal hyperplasia; ^8^ Missing data.

**Table 3 cancers-16-00497-t003:** Radiological and surgical information of the primary and secondary tumor.

	Primary Tumor N = 10	Secondary Tumor N = 10
BI-RADS classification - 1 - 2 - 3 - 4 - 5	0 0 0 0 9	1 9 0 0 0
Surgical treatment breast - Lumpectomy - Mastectomy - Sentinel node - Axillary clearance - No surgery	8 2 6 3 0	0 2 0 0 8
Adjuvant therapy - Chemotherapy - Radiotherapy	2 6	0 1
Neo-adjuvant chemotherapy	5	0

## Data Availability

The data presented in this study are available within the article.

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
