# Peer review of "Breast Cancer in the Tissue of the Contralateral Breast Reduction"

_cancers, 2024, doi:10.3390/cancers16030497_

Round 1
Reviewer 1 Report
Comments and Suggestions for Authors
Major Objection
Patients with previous breast cancer surgery may need contralateral reduction procedure to achieve more symmetry. It is well known that the patients diagnosed with breast cancer have increased risk of developing contralateral breast cancer, and that is the reason why the reduction specimen also must be examined thoroughly.
This is a retrospective cohort study that examined patients who received a symmetrizing breast reduction, after the previous breast cancer treatment in the contralateral breast, between January 2018 and December 2022. The authors introduced the new internal protocol which included excision in one piece, marking, and inking of the specimen on July 20, 2022, with one patient diagnosed with contralateral malignancy after the introduction of the now protocol.
It is well known that the absence of orientation and inking, and removal of breast tissue in multiple pieces cannot provide the accurate status of resection margins and may affect the tumor size evaluation that may affect the patients treatment. Preoperative radiographic evaluation, excision in one piece, marking and inking of the reduction specimen in patients previously diagnosed with breast cancer is mandatory in order to make accurate evaluation and must be a routine practice in every institution that treats breast cancer patients.
Minor objections
The sentence “While surgical intervention remains the primary therapeutic approach for breast cancer, complementary strategies such as chemotherapy, radiation therapy, hormone therapy, and immunotherapy have clinical applications” is not correct because the surgery is not the primary therapeutic approach for patients with aggressive subtypes of breast cancer (triple negative, HER2 positive) larger than 2 cm, and all previously mentioned treatment modalities have clinical impact.
The sentence “Tumor size primarily influences the determination of chemotherapy, targeted therapy, and hormonal therapy” is not completely correct because the hormonal therapy is not influenced by the size of the tumor, it depends only on hormone receptor (ER, PR) expression.
The sentence “Follow-up is indicated for other diagnoses such as lobular carcinoma in situ (LCIS)” is not completely correct because follow up is not the option for pleomorphic type of LCIS for which the treatment is the same as for DCIS.
There is lot of missing data in Table 2.: HER2 status and ER expression for patient 10. and HER2 status for patient 5. in specimen from contralateral breast). Also, HER2 status is not required in the standard pathohistological evaluation in case of in situ carcinoma (patient 1.,2. and 3.). According to ASCO/CAP guidelines percentage and intensity of expression should be noted separately for estrogen and progesterone receptors in both invasive and in situ carcinomas, which is not specified in Table 1.
Comments on the Quality of English LanguageModerate editing of English language is required.
Author Response
Reviewer 1
Major Objection
Patients with previous breast cancer surgery may need contralateral reduction procedure to achieve more symmetry. It is well known that the patients diagnosed with breast cancer have increased risk of developing contralateral breast cancer, and that is the reason why the reduction specimen also must be examined thoroughly.
This is a retrospective cohort study that examined patients who received a symmetrizing breast reduction, after the previous breast cancer treatment in the contralateral breast, between January 2018 and December 2022. The authors introduced the new internal protocol which included excision in one piece, marking, and inking of the specimen on July 20, 2022, with one patient diagnosed with contralateral malignancy after the introduction of the now protocol.
It is well known that the absence of orientation and inking, and removal of breast tissue in multiple pieces cannot provide the accurate status of resection margins and may affect the tumor size evaluation that may affect the patients treatment. Preoperative radiographic evaluation, excision in one piece, marking and inking of the reduction specimen in patients previously diagnosed with breast cancer is mandatory in order to make accurate evaluation and must be a routine practice in every institution that treats breast cancer patients.
- Thank you for your comment. We understand and appreciate the objections. However, we still deem it important to articulate. It is correct that this should be a routine practice in every institution, but this is not yet consistently practiced overall. Therefore, it is crucial to create awareness about this new protocol for more optimal breast cancer treatment.
Minor objections
The sentence “While surgical intervention remains the primary therapeutic approach for breast cancer, complementary strategies such as chemotherapy, radiation therapy, hormone therapy, and immunotherapy have clinical applications” is not correct because the surgery is not the primary therapeutic approach for patients with aggressive subtypes of breast cancer (triple negative, HER2 positive) larger than 2 cm, and all previously mentioned treatment modalities have clinical impact.
- Thanks for your comment. We have corrected it accordingly. See page 2 lines 50-51
The sentence “Tumor size primarily influences the determination of chemotherapy, targeted therapy, and hormonal therapy” is not completely correct because the hormonal therapy is not influenced by the size of the tumor, it depends only on hormone receptor (ER, PR) expression.
- Thanks for your comment. We have corrected it accordingly. See page 3 line 86
The sentence “Follow-up is indicated for other diagnoses such as lobular carcinoma in situ (LCIS)” is not completely correct because follow up is not the option for pleomorphic type of LCIS for which the treatment is the same as for DCIS.
- Thanks for your comment. We have corrected it accordingly. See page 3 line 91
There is lot of missing data in Table 2.: HER2 status and ER expression for patient 10. and HER2 status for patient 5. in specimen from contralateral breast). Also, HER2 status is not required in the standard pathohistological evaluation in case of in situ carcinoma (patient 1.,2. and 3.). According to ASCO/CAP guidelines percentage and intensity of expression should be noted separately for estrogen and progesterone receptors in both invasive and in situ carcinomas, which is not specified in Table 1.
- We have edited the data in Table 2. For the patient number 10 there was missing data. She was treated in another center and we couldn’t retrieve unfortunately her information. For patient 3 the intensity of expression was not available. Thereby, the receptors are solely pertinent to the invasive component; it is not common practice in our institution to assess DCIS based on receptor intensity.

Reviewer 2 Report
Comments and Suggestions for Authors
Reduction mammoplasty of the contralateral breast is a frequent method after lumpectomy for breast cancer. Oncoplastic techniques offer better oncological treatment by allowing more radical removal of the primary tumor and a better aesthetic result and a satisfactory body image of the patient.
The presence of a carcinoma in the "healthy" breast is an important problem as the lack of a pre-operative diagnosis often does not allow the site of the lesion to be precisely identified and therefore is a great challenge for the pathologist and the surgeon as they have no markers for a possible radicalization of the carcinoma.
This condition is crucial for the Multidisciplinary Board to establish the subsequent therapeutic path.
The authors report their interesting experience on a case study which is necessarily limited but which nevertheless offers very precise considerations.
I believe the adoption of a protocol like the one suggested is very useful thanks to the targeted marking of the removed samples of the contralateral breast in order to then perform adequate enlargements or propose a possible mastectomy.
I agree that it is complex to diagnose these lesions with mammography which most of the time, even retrospectively, does not highlight glandular alterations.
The use of MRI could provide more effective elements. I think that in the case of patients who underwent neoadjuvant chemotherapy, the MRI was performed and was negative on the controlateral breast.
I believe that the article is worthy of publication due to its methodological clarity and interest in the topic
Author Response
Reviewer 2
Reduction mammoplasty of the contralateral breast is a frequent method after lumpectomy for breast cancer. Oncoplastic techniques offer better oncological treatment by allowing more radical removal of the primary tumor and a better aesthetic result and a satisfactory body image of the patient.
The presence of a carcinoma in the "healthy" breast is an important problem as the lack of a pre-operative diagnosis often does not allow the site of the lesion to be precisely identified and therefore is a great challenge for the pathologist and the surgeon as they have no markers for a possible radicalization of the carcinoma.
This condition is crucial for the Multidisciplinary Board to establish the subsequent therapeutic path.
The authors report their interesting experience on a case study which is necessarily limited but which nevertheless offers very precise considerations.
I believe the adoption of a protocol like the one suggested is very useful thanks to the targeted marking of the removed samples of the contralateral breast in order to then perform adequate enlargements or propose a possible mastectomy.
I agree that it is complex to diagnose these lesions with mammography which most of the time, even retrospectively, does not highlight glandular alterations.
The use of MRI could provide more effective elements. I think that in the case of patients who underwent neoadjuvant chemotherapy, the MRI was performed and was negative on the controlateral breast.
I believe that the article is worthy of publication due to its methodological clarity and interest in the topic
- Thank you for your review. We appreciate your feedback.
Reviewer 3 Report
Comments and Suggestions for Authors
1. The contribution of the article is unclear, and it is not clear what kind of "protocol" has been provided for follow-up. It is also unclear about the specific steps and implementation methods of this protocol, as described in the abstract.
2. Due to the relatively low number of reported cases in the article compared to the overall patient ratio (10 out of 500 people) and the long time it took for diagnosis to occur (one patient was diagnosed with CBC after 29 years), such a protocol could be considered an excessive diagnosis or overtreatment. In my personal opinion, CBC can be diagnosed during routine follow-up visits with patients.
Comments on the Quality of English LanguageFine
Author Response
Reviewer 3
The contribution of the article is unclear, and it is not clear what kind of "protocol" has been provided for follow-up. It is also unclear about the specific steps and implementation methods of this protocol, as described in the abstract.
- Thanks for your comment. The methods of this protocol are described in paragraphs 2.2 and 2.3 for both surgical and pathological protocol adjustments. Regarding the follow-up, we have corrected it accordingly in paragraph 3.3.3.
- The aim of this study was mainly to create awareness. Before the protocol change the resection were not made in toto, no marking of the resected material and there was no inking process. The pathologist advised changing the protocol after diagnosing numerous CBCs in the breast reduction material in a short period of time. This led to a protocol change for both plastic surgeons and pathologists.
Due to the relatively low number of reported cases in the article compared to the overall patient ratio (10 out of 500 people) and the long time it took for diagnosis to occur (one patient was diagnosed with CBC after 29 years), such a protocol could be considered an excessive diagnosis or overtreatment. In my personal opinion, CBC can be diagnosed during routine follow-up visits with patients.
- We agree that CBC should be diagnosed during routine follow-ups, however the current article refers to a protocol specifically tailored for patients who presented breast cancer in the past and would undergo a contralateral breast reduction. Following the protocol proposed in this article we believe that in case of encountering CBC in the contralateral breast reduction material, the therapeutic options, the size, invasion of the tumor can better be determined. The aim is to establish these practices as routine in all institutions treating breast cancer patients undergoing contralateral breast reduction for optimal patient care. In addition, it emphasizes the need for thorough examination of the breast reduction specimens due to its impact on resection margin accuracy and tumor size evaluation, crucial factors affecting patient treatment. The introduced protocol stresses the significance of excision in one piece, marking, and inking whenever a contralateral breast reduction is requested, for accurate assessments in patients with a history of breast cancer.
Round 2
Reviewer 1 Report
Comments and Suggestions for Authors
Preoperative radiographic evaluation, excision in one piece, marking and inking of the contralaterally breast specimen in patients previously diagnosed with breast cancer should be a routine practice in every institution that treats breast cancer patients.
I support the authors effort to make the improvements in specimen handling protocol in their institution in order to eliminate diagnostic problems and to provide accurate pathohistological report, but this “new protocol” the authors presented in the article (orientation, marking and inking of the resected specimen) is not novelty it is a “common sense” and our obligation as professionals involved in diagnostic of breast cancer since it is well known that the patients diagnosed with breast cancer have increased risk of developing contralateral breast cancer.
The presented study does not provide significant scientific contribution and that is the main reason why I did not support the acceptance in the first place.
Comments on the Quality of English LanguageModerate editing required.
Reviewer 3 Report
Comments and Suggestions for Authors
The author's explanation is reasonable and acceptable.
Such reminders should be promoted.
The relevant protocols have been revised and are easy to understand and follow.